# Are There Conflicts among Energy Security, Energy Equity and Environmental Sustainability in China's Provinces?

**Yijian Ge [1], Lin Liu [2], Xilong Yao [2],\*, Mohammad Aman Honardost [2] and Ujunwa Angela Nwigwe [2]**

[1] KPMG Huazhen Certified Public Accountants, Beijing 100738, China; gexinsheng@tyut.edu.cn
[2] School of Economics and Management, Taiyuan University of Technology, Taiyuan 030024, China; liulin0164@link.tyut.edu.cn (L.L.); aman.honardost013@gmail.com (M.A.H.); sophiaangel229@gmail.com (U.A.N.)
\* Correspondence: yaoxilong@tyut.edu.cn; Tel.: +86-3516014057

**Abstract:** In the process of achieving carbon-peaking and carbon-neutrality goals, conflict situations often arise from advancing energy equity, energy security, and environmental sustainability. Taking China as a case study, we developed an assessment model for conflict levels of energy security, energy equity, and environmental sustainability, based on an evaluation method for the degree of synergy in composite systems, and measured and analyzed the conflict levels of these three dimensions in 2010 and 2017. According to the results, China's overall energy security and energy equity are in a state of conflict. While the level of conflict has eased, the conflict between China's provincial energy security and energy equity is relatively large and more serious in certain provinces, including Shanxi, Heilongjiang, Fujian, Jiangxi, Hubei, Hunan, Chongqing, and Guizhou. Concerning the relationship between energy security and environmental sustainability and between energy equity and environmental sustainability, China as a whole has moved out of a state of conflict, but significant differences exist among different provinces. This paper reveals the relationship between energy security, energy equity, and environmental sustainability in China's energy transition and provides support for the just energy transition of this country.

**Keywords:** energy security; energy equity; environmental sustainability; conflict; China; provincial; energy transition

## 1. Introduction

Many countries have announced ambitious emission reduction targets to accelerate the development of low-carbon energy systems in response to global climate change and increasing greenhouse gas (GHG) emissions. However, in the process of clean-energy and environmental-sustainability development, conflicts between energy security and energy equity have emerged. For example, during Germany's progress toward the goal of carbon emission reduction and the development of renewable energy, renewable electric power accounted for 37.8% of the electricity consumption in this country in 2018 [1]. Renewable energy, such as wind energy and solar energy, has the problems of serious wind and light abandonment, strong random fluctuation and power-system instability due to large-scale grid connection. Therefore, the substantial growth in renewable energy in Germany has not only affected the security of electric power supply, but also aggravated the fluctuation in the electricity price [2], raising concerns about energy equity and energy security. In other countries, the rapid growth in renewable energy has also affected the security of electricity supplies, e.g., the blackout in California and the large-scale power rationing in Hunan in China in 2020. In order to solve the above problems, it may be a good choice to combine technologies that can improve the stability of power grids, such as energy storage. Electric-energy storage has the dual characteristics of power supply and load, which can quickly change the instantaneous power level of the system, and improve the reliability of

power grids and the utilization of renewable energy. Thus, how to effectively coordinate energy security, energy equity, and environmental sustainability requires further study.

The synergy of energy security, energy equity, and environmental sustainability has attracted the attention of many scholars. Specifically, energy security refers to the effective management of a primary energy supply from domestic and external sources, reliability of energy infrastructure, and ability of energy providers to meet current and future demand; energy equity assesses whether an economy can provide abundant, affordable and fairly priced civil and commercial energy; and environmental sustainability represents the transformation process of an economy's energy system to reduce and mitigate environmental hazards and the impact of climate change. The World Energy Council officially named the conflict among these three factors the "energy trilemma" in 2010, constructed the World Energy Trilemma Index (ETI) to quantitatively evaluate the energy performance of different countries, and calculated their overall level of energy security, energy equity, and environmental sustainability [3]. On the basis of ETI, many scholars have contributed to the study of obstacle analysis, goal setting, and policy formulation in the process of energy development in various countries by using principal component analysis, the fuzzy ideal solution method of a regional decision matrix [4], and other methods [5,6]. These scholars have investigated and measured the overall level of energy security, energy equity, and environmental sustainability, which cannot accurately reflect the conflict level among them. However, such accuracy is required to provide more reliable evidence for the formulation of climate policy. To address this problem, this study focuses on the construction and application of a model to calculate the conflict level among the three energy-trilemma components.

Based on ETI, this study constructs an evaluation index system of energy security, energy equity, and environmental sustainability, and establishes an evaluation model of the conflict level of the three dimensions based on the evaluation method of synergy degree in composite systems. China is selected as the study object since it has proposed the target of reaching peak GHG emissions before 2030 and achieving carbon neutrality by 2060; in the process of promoting this goal, China faces a great conflict between GHG-emission reduction and energy security and equity [7,8]. The case study of China can provide a useful reference for other developing countries.

This study contributes to existing literature in two ways. First, we evaluate the energy trilemma from the perspective of the conflict level of its three parts. The perspective of assessment can reveal the dilemma of many countries concerning climate issues regarding the three possible sets of conflicts, namely, between energy security and energy equity, between energy security and environmental sustainability, and between energy equity and environmental sustainability. Second, we establish a comprehensive evaluation model of the three dimensions' conflict level, which can overcome limitations of methods used in the existing literature.

The remainder of the paper is structured as follows: Section 2 contains a literature review related to energy security, energy equity, and environmental sustainability; Section 3 explains the research method and indicator selection; Section 4 reports and discusses the empirical results; and Section 5 summarizes conclusions and potential policy implications.

## 2. Literature Review

The topics of energy security, energy equity, and environmental sustainability have always been important in academic research [4,9]. In the early 1970s, energy security became one of the focal points of international research due to the oil crisis [10–12]. Scholars have analyzed energy security from the perspectives of energy dependence, energy development level, and energy production technology [13,14]. In the 1990s, the climate problem became increasingly prominent. Before and after the signing of the Kyoto Protocol, the topic of solving the environmental-sustainable-development problem of excessive energy consumption has been an important research focus for scholars [15,16]. Energy-efficiency improvement, reduction in fossil-energy consumption, and renewable-energy develop-

ment are considered as the main strategies to achieve environmental sustainability [17,18]. Since the 21st century, many countries have implemented strong policies to reduce GHG emissions, continuously raised energy prices, especially fossil energy prices, and thereby increased the burden on households regarding energy consumption, which has intensified the issue of energy equity [11]. Scholars have discussed the popularization of energy equity from the angle of energy acquisition difficulty and energy affordability [10,19].

In recent years, many countries have proposed their goals regarding carbon peaks and carbon neutralization; implemented carbon taxes, carbon markets, and other policies; and improved the level of environmental sustainability. However, in the meantime, the problems of energy security and energy equity are becoming increasingly serious. The implementation of environmental-sustainability policies raises the energy price to an unreasonable level, which will cause a heavy burden on society, raise concerns about distribution, and result in an energy-equity problem [11]. The academic research on environmental sustainability and energy equity focuses on the interaction mechanism between them. Some scholars have carried out specific research on the relationship among environmental sustainability, energy affordability and energy accessibility [12], and the causal relationship between energy prices and carbon emissions [10]. Other researchers have focused on the conflict between sustainability policies and energy-equity policies [11]. The conflict between environmental sustainability and energy equity is continuously intensifying, because the implementation of environmental-sustainability policies promotes the development of renewable energy power, such as wind power and photovoltaic energy, both of which are characterized by volatile and unstable energy production [20,21] which affects the security of an energy supply and economic benefits [22].

In the research on the relationship among energy security, energy equity, and environmental sustainability, some scholars have comprehensively considered these three dimensions to evaluate energy performance. For example, the World Energy Council established a comprehensive evaluation index system—ETI—for the energy-trilemma components, which measures the overall level in various countries as a base to evaluate the energy performance of these countries [3]. On the basis of the ETI system, some scholars applied interval decision-making matrix, principal component analysis, and fuzzy TOP-SIS analysis to compare and analyze energy performance among different countries [5,6]. By evaluating the overall level of energy security, energy equity, and environmental sustainability, these studies assist in determining influencing factors of energy performance, goal setting, and policy formulation. However, these methods mainly assign different weights to indicators after processing and only consider the comprehensive level of the three dimensions, which cannot reflect the conflict level among them.

To address the shortcomings of the above methods, this study uses a synergy degree model to measure the conflict level of the three components of the energy trilemma. This model can measure the degree of synergy consistency of each element in the process of system evolution and evaluate the conflict level of each element. Therefore, based on ETI, this study analyzes the level of conflict among the three components by using the synergy degree model and then formulates more accurate and reasonable suggestions for the synergistic development of energy security, energy equity, and environmental sustainability.

## 3. Methods and Data

### 3.1. Synergy-Degree Evaluation Model

The evaluation model of the conflict level of China's energy security, energy equity, and environmental sustainability constructed in this study is based on the synergy-degree model, by considering these three components as subsystems [23,24]. The model consists of the following three parts.

#### 3.1.1. Ordering Degree Model of Subsystems

The ordering degree indicates the ordered structure and state of the system. The ordering degree model can identify the variables that play a decisive role in the change

in system state, which are called order parameters. Analyzing the contribution of order parameters to the system, as shown in Formulas (1)–(4), combined with the current situation and problems in the country or region, can help to formulate corresponding solutions and targeted policy recommendations.

To facilitate direct comparison of data, the standardized method is first used to process the data [25]. Let the subsystem be $e_i$ and the order parameter of subsystem be $e_{ij}$, which represents the $j$-th index value of the $i$-th subsystem. With increasing values, the ordering degree of each subsystem will increase. The calculation formula of the ordering degree of subsystem order parameters is as follows:

$$u_i(e_{ij}) = \begin{cases} \frac{e_{ij} - \beta_{ij}}{\alpha_{ij} - \beta_{ij}} & j \in [1, k] \\ \frac{\alpha_{ij} - e_{ij}}{\alpha_{ij} - \beta_{ij}} & j \in [k, n] \end{cases} \tag{1}$$

where $\alpha_{ij}$ and $\beta_{ij}$ are the upper limit and lower limit of the $i$-th system on the $j$-th index, respectively. In Formula (1), the value of $u_i(e_{ij})$ is between 0 and 1, and with increasing values, the contribution to the synergy degree of subsystem $i$ increases.

The "total contribution" of index variables to subsystem synergy can be obtained by integration and determined by the specific structure of the system. In this study, the linear weighting method was used to realize the comprehensive evaluation of the value of $u_i(e_{ij})$ and to obtain subsystem's synergy degree $u_i(e_i)$. The calculation formula is as follows:

$$u_i(e_i) = \sum_{j=1}^{n} w_j u_i(e_{ij}) \tag{2}$$

where $w_j \geq 0$, $\sum_{j=1}^{n} w_j = 1$, and $w_j$ indicates the effect of ordering degree $u_i(e_{ij})$ of the $j$-th index value $e_{ij}$ of subsystem $i$ on the synergy degree $u_i(e_i)$ of subsystem. The CRITIC method is used to calculate the value of $w_j$ [26]. The basic principle of CRITIC method is as follows:

$$\xi_j = \sigma_j \cdot \sum_{i=1}^{n} (1 - r_{ij}) \ (j = 1, 2, \ldots, n) \tag{3}$$

where $\xi_j$ is the influence degree of the $j$-th evaluation index on the system, $\sigma_j$ is the standard deviation of the $j$-th evaluation index, and $r_{ij}$ is the correlation coefficient between the $i$-th and the $j$-th evaluation index. With increasing values of $\xi_j$, the impact of the $j$-th evaluation index on the system and the relative importance of index increases. Therefore, the calculation formula of the objective weight $w_j$ of the $j$-th evaluation index is as follows:

$$w_j = \frac{\xi_j}{\sum_{j=1}^{n} \xi_j} \ (j = 1, 2, \ldots, n) \tag{4}$$

### 3.1.2. The Synergy Degree Model of Each Subsystem

The model measures the degree of conflict among energy-security subsystem, energy-equity subsystem, and environmental-sustainability subsystem.

Assuming that the ordering degrees of the $p$-th subsystem and the $m$-th subsystem are $u_p^0(e_p)$ and $u_m^0(e_m)$ at the base time point and $u_p^1(e_p)$ and $u_m^1(e_m)$ at the observation time point, respectively, the synergy degree between the two different subsystems is calculated as follows:

$$C_{pm} = \lambda \cdot \sqrt{\left[u_p^1(e_p) - u_p^0(e_p)\right]\left[u_m^1(e_m) - u_m^0(e_m)\right]} \tag{5}$$

where $\lambda = \begin{cases} 1, u_p^1(e_p) - u_p^0(e_p) \succ 0 \ and \ u_m^1(e_m) - u_m^0(e_m) \succ 0 \\ -1 \ else \end{cases}, p \neq m.$

In Formula (5), the value of $C_{pm}$ is between $-1$ and 1. With increasing values, the degree of synergy development of the *p*-th subsystem and the *m*-th subsystem increases. The value is determined by both subsystems.

### 3.1.3. The Synergy-Degree Model of Energy Security, Energy Equity and Environmental Sustainability

The model reflects the conflict degree of the three subsystems, namely, energy security, energy equity, and environmental sustainability. Assuming that the ordering degree of the *i*-th subsystem is $u_i^0(e_i)$ at the initial time and $u_i^1(e_i)$ at the observation time, the calculation formula of the synergy-degree index of the whole system is as follows:

$$C = \eta \cdot \sqrt[3]{\prod_{i=1}^{3} [u_i^1(e_i) - u_i^0(e_i)]} \tag{6}$$

where $\eta = \frac{\min[u_i^1(e_i) - u_i^0(e_i)]}{\left|\min[u_i^1(e_i) - u_i^0(e_i)]\right|}$.

In Formula (6), the value of synergy-degree index *C* of the system is between $-1$ and 1. A larger value implies a higher degree of internal integration and synergy of the energy system, which means that the system is in a good running state. In contrast, the whole system is in a poor running state. In Formula (6), the three subsystems interact with each other. If the synergy degree of one subsystem is greatly improved while those of the other subsystems are only slightly improved or even decreased, the whole system is considered in conflict or weak synergy. Only when all three subsystems synchronously increase or decrease, can the whole system be in a synergistic state.

The calculation results of Formulas (5) and (6) reflect whether two or three subsystems promote each other or contradict each other. By analyzing the causes of the contradictions between the two or the three, we can take corresponding measures to coordinate their development and formulate favorable policies.

### 3.2. Selection of Evaluation Index

In this study, the evaluation index system of the synergy degree of energy security, energy equity, and environmental sustainability was established on the basis of ETI formulated by the World Energy Council, and some modifications were made to the index in combination with the actual situation of China. For example, since the purpose of this study is to reveal the internal synergy and conflict among China's energy security, energy equity, and environmental sustainability, the country-context dimension in ETI has been removed. We also referred to the Global Energy Architecture Performance Index Report 2015 issued by the World Economic Forum (WEF) on energy, as well as the research results reported by Matsumoto [27], Kruyt [28], and Yingzhu [29]. Considering the availability of data and the comprehensive coverage of the indicator system, we constructed the conflict-level evaluation index of the three dimensions. Formulas (7)–(9) lay a foundation for calculating the degree of synergy among the three.

### 3.2.1. Energy Security

The energy security dimension reflects the ability of a region to withstand system shocks and meet current and future energy needs. This dimension covers energy-supply security and energy-consumption security. Considering the reality of China, this study uses these two components as secondary indicators, and adds the index of reserve ratio based on the energy-security dimension of ETI. This index is the proportion of the basic regional coal, oil, and natural gas reserves of a particular province in the total reserves of the country, reflecting the storage status of energy and its ability to guarantee economic development [30]. The calculation formula is as follows:

$$X_i = \frac{RC_i + RO_i + RN_i}{\sum\limits_{i=1}^{30} = (RC_i + RO_i + RN_i)} \tag{7}$$

where $X_i$ represents the energy-reserve ratio of the *i*-th province (city, district), ($i = 1, 2, \ldots , 30$); $RC_i$, $RO_i$ and $RN_i$ are the basic reserves of coal, oil, and natural gas of the *i*-th province (city, district), respectively.

### 3.2.2. Energy Equity

The energy equity dimension mainly evaluates whether a region has the ability to provide universally available, reasonably priced, and abundant energy for domestic and commercial uses. The main factors consist of the electrification rate, access to quality energy, and energy affordability, to evaluate energy equity. Referring to the research results of Freitas, an improvement in the electrification rate will lead to an improvement in energy equity [31]. In this study, the electrification rate was added as a secondary index to achieve a more comprehensive energy-equity evaluation index system. The electrification rate is expressed by the per-capita electricity consumption and the proportion of electricity consumption in the total energy consumption [32,33]. High-quality energy access constitutes an important reflection of energy equity and is expressed by the power supply reliability index, which directly reflects the power-supply capacity of enterprises to power users and also the degree to which the power industry meets the power demand of the national economy [34]. The calculation formula is as follows:

$$Y = (1 - Ot/8760) \cdot 100\% \tag{8}$$

where $Y$ is the reliability rate of power supply, $Ot$ is the average annual outage time of users, and 8760 are the hours per year.

### 3.2.3. Environmental Sustainability

The dimension of environmental sustainability mainly represents the ability of a regional energy system to reduce and avoid potential environmental hazards. Resource productivity, decarbonization, and emission and pollution are used to assess environmental sustainability. Based on ETI, we use the line loss rate, coal consumption, and terminal energy intensity to measure resource productivity. The line loss rate is the ratio of power loss to power supply in a certain period, which can comprehensively reflect the power supply economy of a region as well as the technical conditions and management level in a region. It can adequately describe energy resource productivity as a secondary indicator. The calculation formula of line loss rate is as follows:

$$\Delta P\% = (P - P') \div P \times 100\% \tag{9}$$

where $\Delta P$, $P$, and $P'$ represent line loss rate, power supply, and power consumption, respectively.

Coal consumption is the ratio of energy consumed by thermal power plants to output energy and objectively reflects energy efficiency. The combustion of coal and other fossil fuels will produce a large amount of pollutants, such as sulfur dioxide ($SO_2$), nitrogen dioxide ($NO_2$), and particulate matter ($PM_{2.5}$, $PM_{10}$) [35,36]. Aiming at emission and pollution as a secondary index, we added the average annual concentration of $NO_2$ and $SO_2$ as third level indicators on the basis of ETI. Based on the availability of data, methane emission is not included in the index system.

The evaluation model of China's energy security, energy equity, and environmental sustainability conflict level is composed of three measurement levels, and 21 specific indicators. The specific indicators are shown in Table 1. Due to space requirements, the common indicators will not be introduced in detail.

**Table 1.** Evaluation index system of energy security, energy equity and environmental sustainability.

| | First Level Indicators | Secondary Indicators | Third Level Indicators | Unit |
|---|---|---|---|---|
| Evaluation index system of energy security, energy equity and environmental sustainability | Energy security | Energy-supply security | Diversity of primary energy supply | - |
| | | | Reserve ratio | % |
| | | Energy-consumption security | Energy external dependence | % |
| | Energy equity | Electrification rate | Per-capita electricity consumption | kWh/person |
| | | | Proportion of electricity consumption in total energy consumption | % |
| | | Quality energy access | Power-supply reliability | % |
| | | | Electricity price | CNY/kWh |
| | | Energy affordability | Gasoline price | CNY/ton |
| | | | Diesel price | CNY/ton |
| | | | Natural gas price | CNY/m$^3$ |
| | | | Power affordability of residents | |
| | Environmental sustainability | Resource productivity | End energy intensity | - |
| | | | Line loss rate | % |
| | | | Coal consumption | g/kWh |
| | | Decarbonization | Proportion of installed clean-energy-power generation | % |
| | | | Carbon dioxide emission intensity | - |
| | | | Carbon dioxide per capita | tons/person |
| | | Emission and pollution | Annual average concentration of $NO_2$ | μg/m$^3$ |
| | | | Annual average concentration of $SO_2$ | μg/m$^3$ |
| | | | Annual average concentration of $PM_{2.5}$ | μg/m$^3$ |
| | | | Annual average concentration of $PM_{10}$ | μg/m$^3$ |

### 3.2.4. Data Sources

Based on the availability of data, this study used 30 provinces in China, except Taiwan, Hong Kong, Macao, and Tibet, as research objects. The data are based on the statistical data on regional conventional energy in 2010 and 2017, published in *China Statistical Yearbook* (National Bureau of Statistics: http://www.stats.gov.cn/tjsj/ndsj/2011/indexch.htm (accessed on 16 September 2011); http://www.stats.gov.cn/tjsj/ndsj/2018/indexch.htm (accessed on 1 September 2018)), *China Energy Statistics Yearbook*, and *China Electric Power Yearbook*, and wind database (https://wind.in-en.com/ (accessed on 7 May 2022)), China Stock Market Accounting Research (https://www.gtarsc.com/ (accessed on 7 May 2022)), China Price Yearbook, and statistical yearbooks of various provinces (http://tjj.shanxi.gov.cn/tjsj/tjnj/nj2011/html/njcx.htm (accessed on 8 June 2011); http://tjj.shanxi.gov.cn/tjsj/tjnj/nj2018/indexch.htm (accessed on 21 August 2018); etc.). Among the selected indexes, the emission and pollution index is the emission and pollution index of the capital city of each province.

## 4. Results

### 4.1. Analysis of the Conflict between Energy Security and Energy Equity

#### 4.1.1. Overall Analysis

Figure 1 shows the changes in the conflict assessment value between China's energy security and energy equity in 2010 and 2017. In 2010, the conflict between China's overall energy security and energy equity was relatively low. Compared with 2010, China's overall level of conflict between the two dimensions decreased in 2017 but without reaching the level of synergy. The decrease indicates that the conflict between China's energy security and energy equity has been reduced through the efforts of the Chinese government, but without achieving a fundamental change.

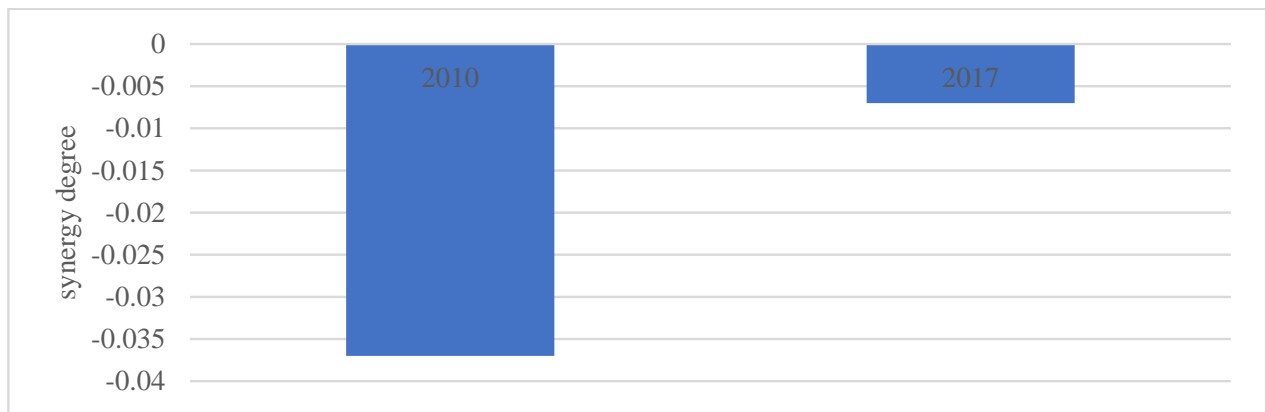

**Figure 1.** Average synergy degree of China's energy security and energy equity.

For example, on the one hand, China has established the role of coal as a "ballast stone" to improve the power safety and supply capacity; on the other hand, China has implemented renewable-energy substitution actions, accelerated the green and low-carbon development of energy through scientific and technological innovation, and improved the regulation capacity of the power system; in addition, China adheres to the priority of saving, significantly promotes the substitution of electric energy, and continues to enact the reform of power marketization, such as the reforming of electricity prices, so as to improve energy equity. However, environmental policies aimed at emission reduction have led to rising energy prices, which make people bear the burden, and the level of electrification varies due to regional differences. Therefore, on the whole, although the conflict relationship between the two has improved, they still have not achieved synergy.

### 4.1.2. Provincial Level

Figure 2 shows the changes in conflicts between energy security and energy equity in China's various provinces in 2010 and 2017. In 2010, energy security and energy equity in most provinces of China were in conflict; only the 10 provinces demonstrate a synergetic development between the two. In 2017, the number of provinces in conflicts between the two dimensions decreased; among them, the relationship between energy security and energy equity in 6 provinces has changed from synergy to conflict, and the relationship in 12 provinces has changed in the reverse.

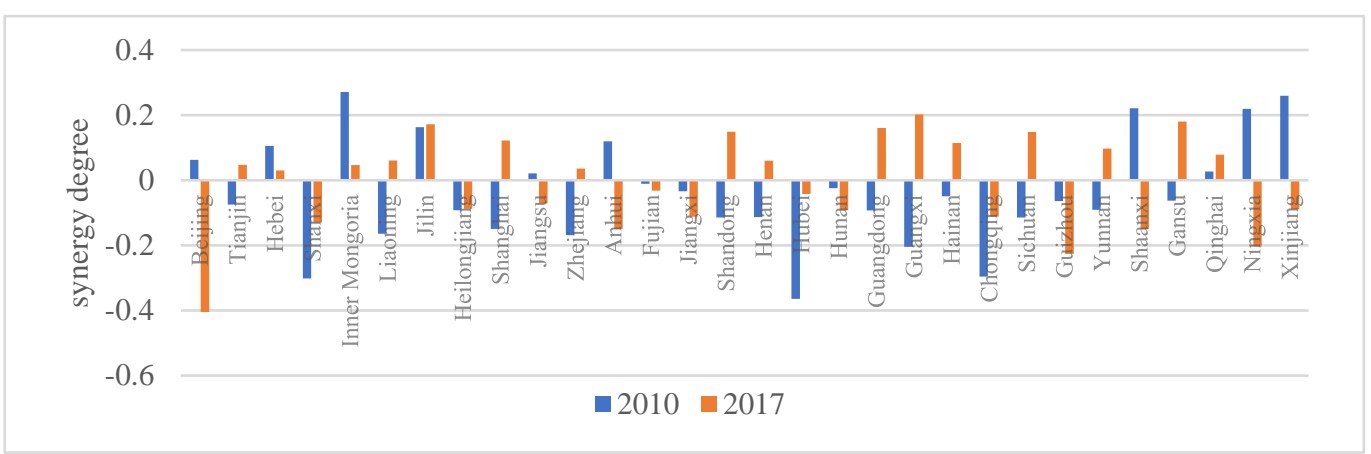

**Figure 2.** Synergy degree of energy security and energy equity in China's provinces.

Among the provinces where energy security and energy equity are in conflict, Fujian, Jiangxi, Hubei, etc., and especially Shanxi, demonstrate a high level of conflict, which has

not been improved since 2010. Shanxi Province is rich in energy and is the national core-power-supply base, but its energy-transformation process is faced with many difficulties, such as the high proportion of coal consumption, the high pressure of dual control of total energy consumption and intensity, a low resource-utilization rate, the restriction of the energy supply, and insufficient development of new energy, which have led to the conflict between energy security and energy equity. However, with the acceleration of Shanxi's energy transformation, the level of conflict between the two has been reduced. In the meantime, a noticeable improvement can be observed for other provinces, such as Shanghai, Shandong, and Guangxi. According to the calculation results, most provinces with high conflict levels are located in the central region of China. The regional differences in basic energy reserves in the central region are large. Most provinces have low energy reserves, cannot achieve energy self-sufficiency, and have poor energy security. In addition, the per-capita power consumption level and the proportion of power consumption in terminal consumption in most provinces are lower than the national average level, and the levels of energy equity and energy security are poor. The majority of provinces fail to achieve synergetic development. Most provinces where energy security and energy equity changed from conflict to coordination are located in the eastern part of China. The eastern part of China is located in the coastal area, most provinces are economically developed, and the level of electrification and burden on households is relatively high; however, the demand for energy is large, and the security of supply is generally low. In recent years, as the eastern region has supported the development of new energy, improving the diversity of energy supply, and ensuring a higher energy self-sufficiency rate, the conflict between energy security and energy equity in the eastern region has improved with the enhancement of energy security.

Among the provinces demonstrating synergetic development of energy security and energy equity, Inner Mongolia, Jilin, Hebei, and Qinghai exhibit a good synergistic state of the two dimensions. The other provinces, such as Beijing, Ningxia and Anhui, have dropped their energy security and energy equity relationship from synergetic development to conflict. The results show that most provinces where the association between these two dimensions has changed to conflict are located in the western region of China. The energy reserves in this region are the highest in the country, and most of the provinces can achieve energy self-sufficiency, which ensures the energy-supply security of the western region to a large extent. However, the overall economic development of this region is poor. During the study period, the natural growth rate of the average population is much higher than the national average, and the per-capita energy consumption level is relatively low. In addition, the electrification rate and household-affordability rate of the energy prices, as well as the degree of energy equity, are generally poor, which leads to the conflict between energy equity and energy security.

### 4.2. Analysis of the Conflict between Energy Security and Environmental Sustainability

4.2.1. Overall Analysis

Figure 3 shows the changes in the conflict between China's energy security and environmental sustainability in 2010 and 2017. For the country, the synergy degree between the two dimensions increases from 2010 to 2017. In both years, the relationship between energy security and environmental sustainability shows synergetic development. This shows that China's energy security and environmental sustainability are in a good relationship of mutual promotion, and the overall direction of development points toward a higher degree of synergy.

The reason for this is that, in terms of energy supply, on the one hand, China reduced the use of fossil energy, and vigorously developed renewable energy and clean energy; on the other hand, it constructed a diversified energy supply system to reduce external dependence. In terms of energy demand, China strives to save energy, improve energy efficiency and implement a large-scale demand response; in terms of energy technology, China has vigorously promoted the flexible transformation of coal motor units, accelerated

the development and utilization of CCUS technology, and strengthened the research and development of key technologies for renewable-energy consumption, such as energy storage technology; in terms of energy cooperation, China promotes the international new energy cooperation mechanism and has actively built the international energy Internet. The above measures not only guarantee energy security, but also effectively reduce environmental pollution and promote the sustainable development of the environment.

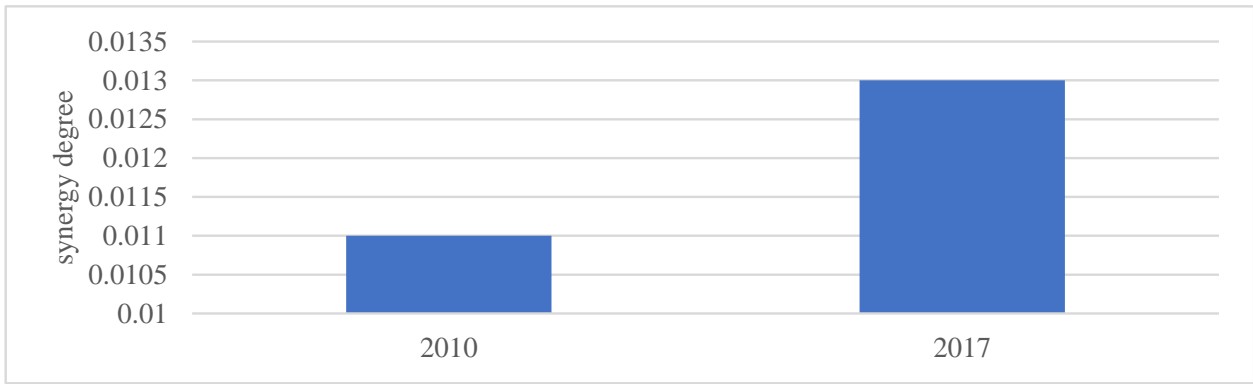

**Figure 3.** Average synergy degree of energy security and environmental sustainability in China.

4.2.2. Provincial Level

Figure 4 shows the changes in the conflict between energy security and environmental sustainability in China's provinces in 2010 and 2017. In 2010, the two dimensions are in conflict in more than half of China's provinces; the remaining provinces are in a state of good synergy, such as Beijing and Tianjin. In 2017, the number of provinces in which the two are in synergy increased; among them, the relationship between energy security and energy equity in six provinces has changed from synergy to conflict, and the relationship in eight provinces has changed in the reverse.

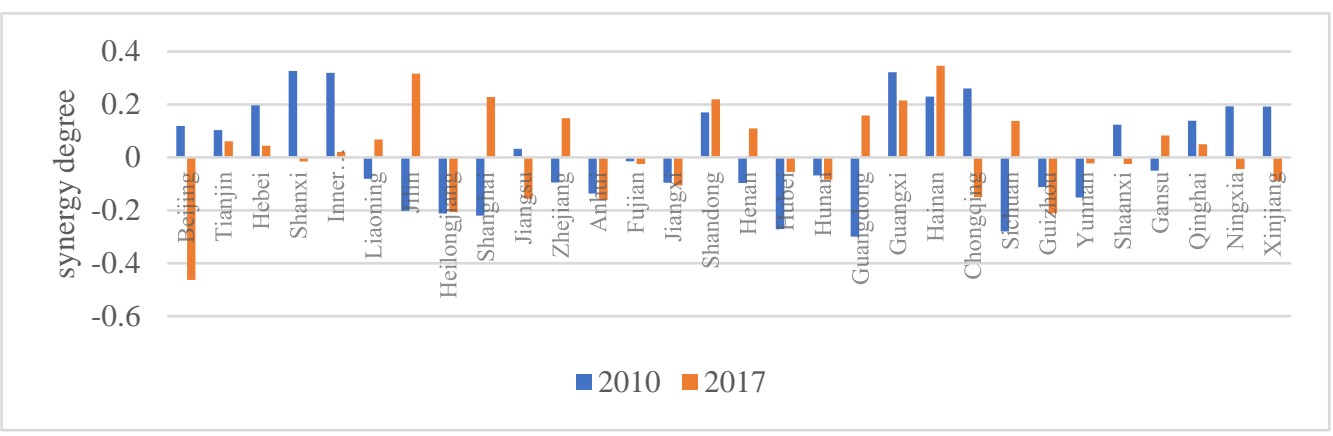

**Figure 4.** Synergy degree of energy security and environmental sustainability in China's provinces.

Among the provinces where energy security and environmental sustainability are in conflict, Anhui, Fujian, Jiangxi, etc., demonstrate a higher conflict level without any improvement. Compared with other provinces, Heilongjiang and Hubei have a higher overall level of conflict. Heilongjiang is the province with the richest energy reserves among the three northeastern provinces, and has made great contributions to national economic development in terms of energy security. However, the over exploitation of energy has brought huge environmental pressure to Heilongjiang, such as the serious pollution of the atmosphere by methane gas produced in the process of exploitation. Moreover, its energy

consumption mainly depends on traditional fossil energy, and a large number of harmful substances such as carbon dioxide and sulfur dioxide are emitted in the process of use, resulting in serious environmental pollution. For Hubei, it has low energy reserves, and, in the process of development, it has taken over the transfer of enterprises with high energy consumption and high pollution in the eastern developed region. The level of environmental sustainability and energy security is relatively poor, and an overall state of synergetic development has not been achieved. Provinces such as Liaoning, Jilin and Shanghai have changed their energy security and environmental sustainability from conflict to synergy. Most provinces where the conflict between energy security and environmental sustainability has improved significantly are located in the eastern region of China. The eastern region is the fastest-growing region in China. These regions generally lack fossil-energy resources. Therefore, they continue to increase investment in wind energy, solar energy, biomass energy, nuclear energy, and other new energies. During economic development, higher investments have been made in environmental protection, which ensures the coordinated development of environmental sustainability and energy security.

Among the provinces with the synergetic development of energy security and environmental sustainability, Inner Mongolia, Guangxi, Hainan, etc., are the provinces with a good synergistic relationship between the two dimensions. The relationship between energy security and environmental sustainability in other provinces is developing in the direction of conflict, including in Shanxi, Chongqing, and so on. The results show that the provinces with a co-promotion relationship between energy security and environmental sustainability are mainly distributed in the eastern and western regions of China, and the synergetic development is worse in the central region than in the eastern and western regions. Most provinces where the relationship between these two energy-trilemma components is developing in the direction of conflict are located in the western region. Limited by economic level and although the western region is rich in resources, its industrial technology as a whole is relatively backward, causing serious waste and environmental pollution in the process of mining and processing. The mode of economic development reflects the characteristics of extensive high pollution and low efficiency and relies too much on coal. In addition, energy intensity is too high, environmental sustainability continues to deteriorate with industrial development, and the conflict with energy security is relatively serious.

*4.3. Analysis of the Conflict between Energy Equity and Environmental Sustainability*

4.3.1. Overall Analysis

Figure 5 shows the changes in the conflict between energy equity and environmental sustainability in China in 2010 and 2017. Although the synergy degree of energy equity and environmental sustainability in China is low, it shows an overall growth trend. In 2010 and 2017, the relationship between the two dimensions demonstrates a good mutual promotion, and the overall development is more synergetic.

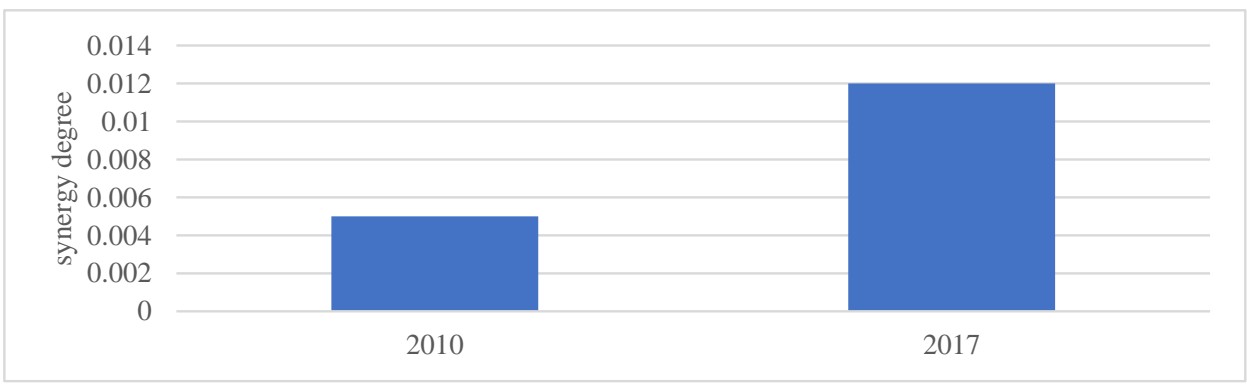

**Figure 5.** Average synergy degree of energy equity and environmental sustainability in China.

This is because, with the passage of time, electrification in the new era emphasizes the transformation of power supply structures to green and low-carbon on the supply side, the improvement in energy and power utilization efficiency on the consumption side, and the universality and benefits of power services on the level of sustainable development, so as to meet the power demand of the people for a better life. In other words, while improving energy equity, environmental sustainable development can be promoted. During this period, initiatives such as clean heating and switching from coal to electricity in China have accelerated the level of electrification, so energy security and environmental sustainability are moving in a more synergetic direction.

4.3.2. Provincial Level

Figure 6 shows the changes in the conflict between energy equity and environmental sustainability in China's provinces in 2010 and 2017. In 2010, thirteen provinces faced a conflict relationship between the two dimensions. In 2017, the energy equity and environmental sustainability in other provinces are in a state of synergistic development, except for Yunnan Province, which has transformed from synergy to conflict.

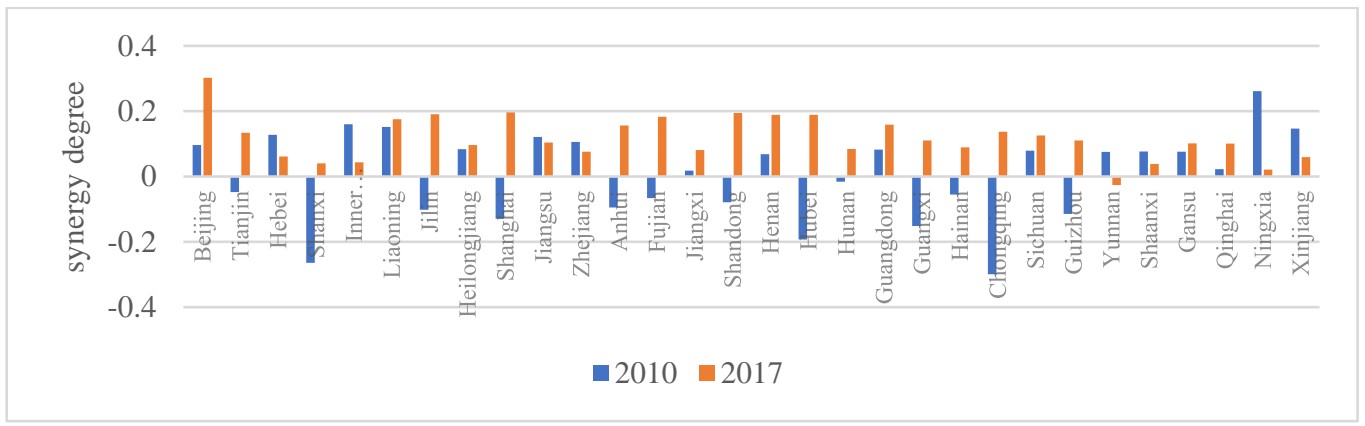

**Figure 6.** Synergy degree of energy equity and environmental sustainability in China's provinces.

All provinces that were in conflict between energy equity and environmental sustainability changed to a synergetic development state in the investigated period. For example, in Tianjin, Shanxi and Jilin, the synergy degree between the two dimensions increases in the investigated period, and the relationship changes from conflict to synergy. During the investigated period, the energy equity and environmental sustainability of eastern and western provinces are developing toward a higher degree of synergy. The reason for this is that most provinces are economically developed, the electrification level and energy-bearing level of residents are relatively higher, and attention has been paid to the investment in clean energy. Meanwhile, the central provinces are changing from a conflict state to a synergistic state, especially in Chongqing. The reason for this is that, from the perspective of economic structure, Chongqing's economy has developed rapidly, and in the process of steadily developing the secondary industry from 2010 to 2017, it has gradually adjusted the industrial structure and accelerated the development of tertiary industry, which is conducive to improving energy equity; from the perspective of energy consumption, Chongqing's energy consumption is dominated by coal, and its economic development depends heavily on coal; from 2010 to 1017, Chongqing gradually paid attention to low-carbon economic development and strengthened the use of green energy. The proportion of coal consumption showed a downward trend; in terms of carbon emissions, the growth rate of carbon emissions slowed down. Especially after 2015, the total carbon emissions showed a downward trend for the first time.

Among the provinces where energy equity and environmental sustainability are in a relationship of synergetic development, Ningxia, Hebei, Beijing, etc., have been in a good

synergistic state. The synergy degree is relatively high in Beijing and Liaoning Province. For Beijing, its green-technology innovation is in a leading position, and has a strong foundation of manufacturing and producer services and high research and development investment. For Liaoning, as a key province of revitalizing the old northeast industrial base, it has taken many measures in the clean development of the power grid. For example, in terms of clean-energy consumption, it uses market-oriented means to allocate peak-shaving resources and actively promote the development and utilization of new peak-shaving technologies; in terms of electric-energy substitution, in response to the national new energy-vehicle policy, the promotion and application of electric boiler, heat pump and other electric technologies have been accelerated. Yunnan Province is the only province where energy security and environmental sustainability are transforming to a conflict state. During the investigated period, Yunnan achieved rapid socio-economic development, improved its electrification rate and household affordability rate, and improved energy equity. However, due to the rapid industrialization and urbanization process as well as unreasonable resource development and extensive energy utilization, this province is facing a new challenge. It is difficult to maintain a good relationship between environmental sustainability and energy equity, which turned from a synergetic to conflict state.

### 4.4. Analysis of the Conflict among Energy Security, Energy Equity and Environmental Sustainability

#### 4.4.1. Overall Analysis

Figure 7 shows the overall conflict change of China's energy security, energy equity, and environmental sustainability system in 2010 and 2017. In 2010, China's overall trilemma system was in a conflict state. In 2017, the conflict relationship between the three dimensions has been improved in China and transformed into a good synergistic relationship. This change demonstrates that, because of the effective implementation of China's energy policy and the efforts of the government, the conflict between China's energy security, energy equity, and environmental sustainability has been transformed fundamentally, into a good promoting relationship.

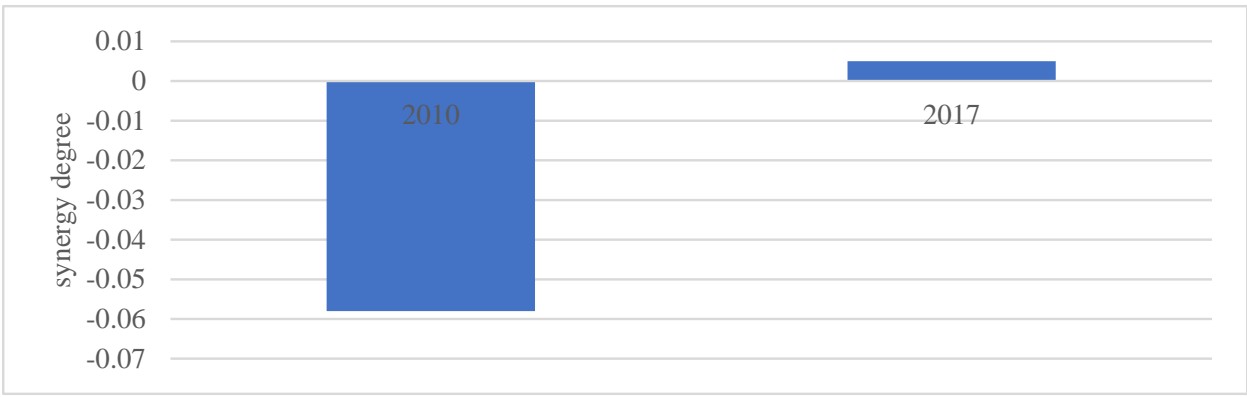

**Figure 7.** Average synergy degree of energy equity, energy security and environmental sustainability in China.

For example, in terms of energy security, China has effectively played the basic regulatory role of coal power, continuously improved oil and gas exploration and development, continuously improved the production, supply, storage and marketing system, and improved the power security and supply capacity; in terms of energy equity, it has promoted the large-scale development and utilization of clean energy and deepened the market-oriented reform of electricity prices; in terms of environmental sustainability, it has adjusted the urban energy structure, gradually replaced coal with gas and heat, taken energy-saving measures, and developed clean alternative energy. As a result, the relationship between the three has changed fundamentally.

### 4.4.2. Provincial Level

Figure 8 shows the changes in the conflict among energy security, energy equity and environmental sustainability in China's provinces in 2010 and 2017. In 2010, most provinces experienced a state of conflict regarding the relationship of the three. In 2017, the number of provinces in synergy regarding all dimensions increased to 15; among them, the relationship between energy security and energy equity in 5 provinces has changed from synergy to conflict, and the relationship in 12 provinces has changed in the reverse.

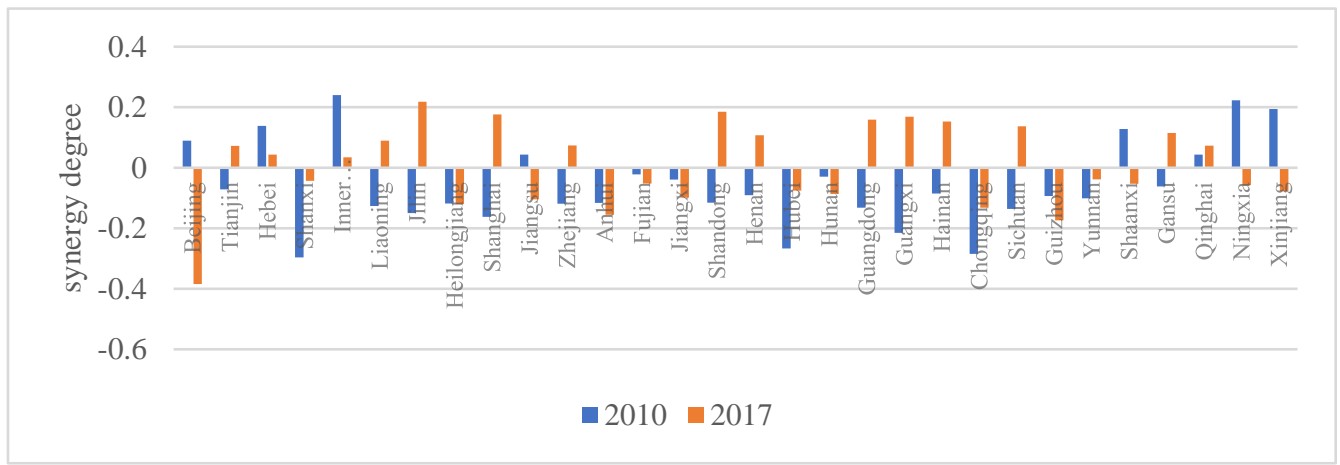

**Figure 8.** Synergy degree of energy equity, energy security and environmental sustainability in China's provinces.

Among the provinces in a state of conflict, Shanxi, Heilongjiang, Anhui, etc., show a relatively high level of conflict and have not improved, especially Hubei and Chongqing, which continue to be in a serious conflict state. For Chongqing, its energy resources are not rich, and its energy consumption is dominated by fossil energy. Although it includes a natural-gas-rich mining area, with the rapid development of Chongqing's economy, the demand for natural gas also increased sharply, resulting in high external dependence, high energy demand and energy-security problems. For Hubei, fossil energy is scarce and its external dependence on energy is too high. Although it is a province with significant hydropower, it has changed greatly under the influence of climate and reservoir operation, and the electricity price is higher than the national average level, so that the phenomenon of seasonal power shortages in Hubei cannot be fully alleviated. Therefore, the degree of energy equity and energy security is consistently low. However, the conflict situation in other provinces has improved and transformed into synergetic development relations, such as in Tianjin, Liaoning and Jilin. In addition, according to the calculation results, most provinces where energy security, energy equity, and environmental sustainability have changed from a conflict relationship to a good promotion relationship are located in eastern China. These provinces enjoy abundant renewable energy and nuclear-energy resources with higher investments in rapid economic development, environmental protection, energy saving, and advanced energy technology to ensure reasonable energy prices and energy efficiency. Significant progress has been made in energy security, energy equity, and environmental sustainability.

Among the provinces where the three dimensions are in a synergetic development relationship, Hebei, Inner Mongolia, and Qinghai have always maintained a good synergy. Other provinces, including Beijing, Jiangsu, Shaanxi, Ningxia, and Xinjiang, change from mutual promotion to conflict. Among them, Shaanxi, Ningxia, and Xinjiang are in the western region. While the average degree of synergy in the western region is negative during the study period, a general development toward the direction of conflict mitigation can be noted. Although the western region has a certain amount of resource reserves, most of the provinces are economically undeveloped with insufficient investment in environ-

mental protection, low energy-technology efficiency, and high energy and carbon emission intensity. In addition to a high degree of energy security, energy equity and environmental sustainability are generally poor compared with other regions.

## 5. Conclusions and Policy Implication

### 5.1. Conclusions

Based on the framework of the Energy Trilemma Index (ETI), this study established a conflict-level evaluation model of energy security, energy equity, and environmental sustainability from the conflict relationship among these three dimensions and evaluated the conflict level in 30 provinces of China in 2010 and 2017. The conclusions are as follows:

First, China's overall energy security and energy equity are in a state of conflict, but the level of conflict has declined. Compared with 2010, the number of China's provinces experiencing energy security and energy equity conflict has significantly decreased in 2017. The conflict between these two dimensions is more serious in most provinces of the central and western regions of China, while it is less pronounced in most provinces of eastern China and has changed to the direction of coordination.

Second, China's overall energy security and environmental sustainability have entered a good state of coordinated development. Compared with 2010, the number of provinces in conflict between energy security and environmental sustainability in China decreased in 2017, especially in most provinces in the eastern and western regions of China. However, the conflict between these two dimensions in most provinces of the central region is still serious.

Third, China's overall energy equity and environmental sustainability have moved from a conflict state to weak collaborative state. Compared with 2010, except for Yunnan Province, energy equity and environmental sustainability in the vast majority of China's provinces in 2017 have moved out of the state of conflict and gradually stepped into the state of coordinated development. Most provinces in the eastern and western regions of China are developing toward a more coordinated relationship state of energy equity and environmental sustainability.

Fourth, the conflict among energy security, energy equity, and environmental sustainability in China has been fundamentally transformed and has entered a state of coordinated development. The number of provinces in conflict between all three dimensions has decreased significantly, and most of them have been transformed into a relationship of coordinated development. Most provinces where this conflict has been significantly improved are located in the eastern region of China, while most provinces in the western region remain in a state of conflict.

### 5.2. Policy Implication

Some important policy implications can be deduced from the findings of this study. First, the Chinese government needs to attend to the conflict between energy security and energy equity by implementing measures such as appropriate subsidies for energy storage at both supply and demand sides and reducing the burden ratio of power users to improve the energy equity of the majority of residents. Second, the Chinese government should accelerate the promotion of the coordinated development level of energy equity and environmental sustainability, introduce subsidies to support low-income residents to participate in photovoltaic rooftop electricity generation, and encourage these low-income residents to use renewable-energy power. Third, central China should implement measures to reduce the conflict between energy security and environmental sustainability, such as moderately developing renewable-energy power projects combined with energy storage and reducing the proportion of coal-based energy in the supply of electrical power.



**Author Contributions:** Conceptualization, Y.G.; methodology, Y.G.; software, L.L.; validation, U.A.N. and M.A.H.; formal analysis, U.A.N. and X.Y.; investigation, M.A.H.; data curation, L.L. and U.A.N.; writing—original draft preparation, Y.G. and X.Y.; writing—review and editing, Y.G. and L.L.; visualization, M.A.H.; supervision, X.Y. All authors have read and agreed to the published version of the manuscript.

**Funding:** This research received no external funding.

**Institutional Review Board Statement:** Not applicable.

**Informed Consent Statement:** Not applicable.

**Data Availability Statement:** Not applicable.

**Acknowledgments:** We acknowledge the financial support from the Program for the Top Young Academic Leaders of Higher Learning Institutions of Shanxi, the Program for the Philosophy and Social Sciences Research of Higher Learning Institutions of Shanxi, and the Qualified Personnel Foundation of Taiyuan University of Technology.

**Conflicts of Interest:** The authors declare no conflict of interest.

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
