# Peer review of "Are There Conflicts among Energy Security, Energy Equity and Environmental Sustainability in China’s Provinces?"

_sustainability, doi:10.3390/su14116873_

Round 1

Reviewer 1 Report

The manuscript presents a conflict level of energy security, energy equity, and environmental sustainability based on the conflict relationship between these three dimensions, and evaluates the conflict level in 30 provinces in China in 2010 and 2017. It is an interesting case study and the authors have documented it well. The technical contribution is significant. 

Reviewer 2 Report

In this review, the author performed a sample analysis on the energy sustainability based on the already available data from the literature. The overall structure of the work is average and may require additional clarifications/justifications before being accepted for publication. Detail below to name a few:

a) The organization of the work is below par and I do not understand how the methods described through equations 1-9 reveal in determining energy policy.

b) The Energy security and Energy equity concepts have to be justified with additional sets of real life data.

c) One major drawback of the present work is that the author only focuses the analysis from a few regions in a country. My question is, what about the global perspective?

d) Towards the above point, a few case studies and associated comparison must be appended in the revised manuscript.

e) The qualities of the figures are below par and must be improved to meet international standard.

f) The manuscript must be checked for possible errors in the English language use. 

Reviewer 3 Report

The manuscript reported the assessment model for conflict levels of energy security, energy equity, and environmental sustainability in various provinces of China. It is based on an evaluation method for the degree of synergy in composite systems, and the conflict levels of these three dimensions are measured and analyzed. It is found that China's overall energy security and energy equity are in a state of conflict. While the level of conflict has eased, great differences exist among different provinces. This paper may provide support for the just energy transition of this country.

I consider the content of this manuscript will definitely meet the reading interests of the readers of the Sustainability journal. However, there are certain English spelling and grammar issues, and also the discussion and explanation should be further improved.

Therefore, I suggest giving a minor revision and the authors need to clarify some issues or supply some more experimental data to enrich the content. This could be a comprehensive and meaningful work after revision.

1. For grammar issues, it is suggested that the author double-check the small grammar errors in the full text, especially the lack and redundant use of definite articles. I will only point out some of them here, of course, not all of them. For example:

Page 4, With the increasing value of ξj, the impact of the j-th evaluation index on the system

Page 5, In contrast, small values indicate a serious internal conflict in the system, If the synergy degree of one subsystem is greatly improved while those of the other subsystems are only slightly improved or even decreased;

Page 8, Figure 2 shows the changes in conflicts between the two dimensions, which are not in conflict and are positioned at the level of synergy, most provinces with high conflict levels are located in the central region of China. The regional differences in basic energy reserves in the central region are large, and so on.

2. For the Keywords, ‘conflict’, ‘China’, ‘provincial’, and ‘energy transition’ should be added in order to attract a broader readership.

3. Page 1, The substantial increase in renewable energy in Germany not only affects the security... but also aggravates the fluctuation of the electricity price[2], raising the concern of energy equity and energy security and the rapid growth of renewable energy has also affected the security of electricity supply.

It seems that energy security and energy equity are largely related to the rapid development of renewable energy. But what is the detailed mechanism of this issue? It should be further explained.

For example, Renewable energy sources such as solar energy and wind energy are unstable and intermittent during generation, and thus these valuable electric energies are difficult to apply continuously and stably. To tackle this issue, the employment of large-scale energy storage systems combined with renewable energy may greatly improve the utilization rate and stability of renewable energy’ [ChemSusChem 15.1 (2022): e202101798]. The drawbacks of renewable energy should be mentioned, and since batteries are also developing fast in China, will the energy storage be helpful to combine with renewable energy sources to solve the problem of energy security and equity problems? This may be an interesting point, since ‘electric energy storage may enhance the quality and reliability of the electrical grid, increase the utilization of renewable resources, and enhance the flexibility of the integration of sustainable energy into the power system’ [IEEE Transactions on Smart Grid 6.5 (2015): 2395-2402; Journal of Power Sources 493 (2021): 229445].

4. Page 5, Considering the availability of data and the comprehensive coverage of the indicator system, we construct the conflict level evaluation index of the three dimensions. Where is the data adopted in this manuscript from? This issue should be introduced briefly. Page 7, ‘3.2.4. Data Sources’, can this data be downloaded online? I suggest providing some references or the website/link for downloading the data as the main resources for the modelling.

5. Page 7, The decrease indicates that the conflict between China's energy security and energy equity has been reduced through the efforts of the Chinese government, but without achieving a fundamental change. Since the conflict has been reduced, which are the detailed efforts of the Chinese government? At least, some of the efforts should be mentioned briefly to be a guideline for the readers and other governments as a benchmark/reference.

6. I think the text is too detailed to list almost every province in China. I suggest selecting only the most representative provinces to explain the problem, not all. This is especially like a boring list of provinces, and it appears too many times in the article.

7. Although the model established through the ETI framework has well described the three-dimensional relationship between the three factors, I think there are few energy policies integrating theory with practice. The example of Yunnan Province specifically illustrates what policies it adopts and what trends the data lead to. Such an expression and interpretation method is more worthy of praise. I suggest that the article should be combined with actual policies so that the data and policies correspond one by one, which is more persuasive and referential  [Energy Policy 67 (2014): 595-604.].

Reviewer 4 Report

The article in content and methodology is very interesting. However, there are some issueswith the sound and clarity of the general concepts underlying the research.

The introduction is extremely repetitive and generic. The phrase "energy security, energy equity, and environmental sustainability" appears dozens of times in the first two pages without any explanation of what these concepts refer to. It is therefore advisable to introduce these key concepts for the article already in the introduction by giving a brief definition of them.

The presentation of the results is limited to a description of them with very few reflections on why certain provinces are disadvantaged compared to others. The discussion of the results should be expanded by going more into the substance of the issues that generate certain differences.

The tone of the article is consequently a little too condescending towards the policies currently in place in the country and a little timid in identifying the reasons for certain disparities.

Round 2

Reviewer 2 Report

The revised version may be considered for publication.